# The Effects of Open Innovation at the Network Level

**Lu Cheng**

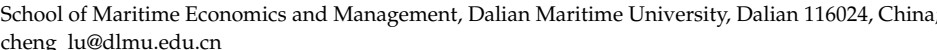

School of Maritime Economics and Management, Dalian Maritime University, Dalian 116024, China; cheng_lu@dlmu.edu.cn

**Abstract:** To open the black box of how open innovation works at the network level, we particularly focused on the effects of TFs' collective openness of external knowledge search on RIN innovation performance in different innovation environments of disruptiveness. To conduct the research, we adopted a bottom-up research approach and designed an agent-based simulation model. The simulation results show that either collective breadth or depth exerts significant effects on the RIN innovation performance, and their effectiveness is significantly moderated by disruptiveness. Our results reveal the followings: (1) RIN innovation performance can be considerably enhanced by high collective openness, but it is not necessarily true that more collective openness is better, which reflects that the "inverted U-shaped" relationships broadly argued in firm-level open innovation studies also exists at the network level. (2) The effect of collective openness depends on the disruptiveness of innovation environment. The likelihood of a positive effect of collective openness on RIN innovation performance increases as the disruptiveness is enhanced. The trends of the polarization of individual TFs' innovation performance in an RIN is alleviated as the disruptiveness is enhanced. Based on these findings, we give some guidance of innovation policymaking. When the industry is in its emerging stage, aggressive collective openness—high collective breadth and depth—aiming at achieving high RIN innovation performance is recommended. When the industry steps into its developing stage, directed collective openness—high collective depth and moderate collective breadth—aiming at fostering future industry leaders in the RIN is recommended. When the industry reaches its maturity, conservative collective openness—low collective depth and moderate collective breadth—aiming at maximum utilization of current RIN's competitive advantages is recommended.

**Keywords:** open innovation; collective openness; RIN innovation performance

## 1. Introduction

In the context of knowledge explosion and globalization, open innovation (OI), which stresses "the use of purposive inflows and outflows of knowledge to accelerate internal innovation, and expand the markets for external use of innovation, respectively" [1], has become a worldwide economic phenomenon in many industries. Millions of technology-based firms (TFs) innovate through a complex set of interactions or collaborations with external actors, e.g., suppliers, customers, research institutes, and even competitors [2–4]. Accordingly, the widespread implementation of OI subverts individual TFs' traditional innovation pattern from being self-dependent to being interdependent [5–7] and simultaneously accelerates the emergence and evolution of regional innovation networks (RINs) [8–10]. It has been widely accepted that OI not only plays an important role in an individual TF's survival and growth but also exerts a significant effect on an RIN's evolution, which reflects the sustainability of regional economic development [11–13]. Therefore, OI has become a core component in both firm innovation strategy and government innovation policy.

However, compared with to firm OI strategy setting, research on OI offers fewer theoretical implications to government OI policymaking [14,15] because the majority of OI studies are firm-level or company-centric, e.g., focusing on how individual TFs' OI strategies and practices influence their own innovation performance [16,17]. Consequently,

prescriptions drawn from these works are suitable for individual TFs, but the availability and applicability to RINs are not guaranteed. Actually, the ultimate goal of OI implementation in an RIN often conflicts with that in a TF, i.e., collective optimization versus individual optimization [18,19]. This conflict indicates that the logic of government innovation policy-making is quite different from that of firm innovation strategy setting.

Network-level analysis of OI is the prerequisite and foundation to provide useful insights on innovation policymaking for OI, and the issue of network-level innovation performance ought to be considered first. Similar to those of firm innovation strategy, the primary motivation and the ultimate goal of government innovation policy are innovation performance improvement—higher regional innovation capacity and higher innovative outcomes [20,21]. In turn, network-level innovation performance is the most important barometer for innovation policymakers. Nevertheless, the research on this issue is still in its infancy [18,22–24]. In contrast to the relatively mature research on firm-level innovation performance, the studies of network-level innovation performance are fragmented in several scatter fields, such as supply chain networks, policy-driven networks, collaborative networks, and business networks. They are not built on each other and have not come to an agreement on network performance conceptualization and measurement.

In this paper, we aim to open the black box of OI effects at the network level. Concretely, we focus on the relationship between collective OI practice and RIN innovation performance and determine how it is moderated by the disruptiveness of industrial innovation, which is an important narrow sense of RIN environment. First, we confined OI practice to the typical OI behavior—external search for knowledge—and use the notion of "collective openness" to describe collective OI practice as well as the research of individual firm OI. We identify collective openness in terms of the breadth and depth dimensions. Then, we develop an assessment pattern for RIN innovation performance, in which we use the mean and variance to measure an RIN's innovation performance from the perspective of knowledge learning and creation. Next, we adopt the agent-based modeling and simulation method (ABMS) to conduct our bottom-up research. In the end, we analyze and discuss the results of simulation experiments that show the effects of collective openness on RIN innovation performance under different kinds of disruptiveness.

We organize the rest of our paper as follows: in Section 2, we briefly review the relevant key concepts of external search for knowledge and network performance and develop the research model based on the behavioral process of individual TFs' external knowledge search. In Section 3, we provide the basic descriptions and mathematical abstractions of the agent-based simulation model and set the details of the numerical simulation experiments. In Section 4, we present the simulation results that show the effects of collective openness on RIN innovation performance in different innovation environments within different disruptiveness degree. In Section 5, we discuss the managerial implications of the results, which may offer useful insights on innovation policymaking.

## 2. Theoretical Background

### 2.1. External Knowledge Search

#### 2.1.1. The Openness of External Knowledge Search

OI practice contains many kinds of sub-activities, such as external knowledge search, outsourcing, crowdsourcing, licensing, etc. [1,6,7]. External knowledge search, through which individual TFs acquire complementary knowledge and novel ideas from external sources to stimulate innovation, is one of the most typical OI behaviors and has attracted a great deal of attention and has been frequently discussed in the existing OI literature. Among relevant studies, the vast majority focus on the topic of the openness of individual TFs' external knowledge search (individual openness for short) and its effects on TFs' innovation performance [6,7,25–27].

A bidimensional framework established by Laursen and Salter (2006) [25] is widely accepted in the individual openness analysis. In their framework, individual openness is discerned from the dimensions of breadth and depth. Breadth reflects how widely a

TF searches for knowledge from outside, while depth reflects to what extent a TF utilizes outside knowledge. They also provide a feasible measure method for individual openness: a TF's breadth is evaluated as its total number of external sources or channels, and depth is evaluated as the number of deep interactions with external sources or channels.

Contrary to the consensus of the dimensions and measurement of individual openness, scholars hold different opinions on the effectiveness of individual openness. For instance, through a large-scale survey of 2707 U.K. manufacturing firms, Laursen and Salter (2006) found that breadth and depth have a curvilinear (an inverted U-shape) relationship with firm innovation performance [25]. Arruda, Rossi, Mendes, et al.'s (2013) survey on 72 Brazilian firms [28] and Kobarg, Stumpf-Wollersheim, and Welpe's (2019) investigation on 218 innovation project in German [29] also reached the same conclusion. In contrast, based on a questionnaire investigation of TFs in Zhejiang province, China, Chen, Chen, and Vanhaverbeke (2011) proved that both breadth and depth are positively related to TF's innovation performance and argued that the decreasing returns of external search strategies (inverted-U shaped relationships) are not always present [30]. These findings are supported by Ahn, Minshall, and Mortara's (2015) empirical research on 306 Korean innovative SMEs [31]. In addition, by investigating 500 Italian TFs, Martini et al. (2012) contradicted the "inverted U-shaped" relationship between individual openness and firm's innovation performance. Their research shows that depth has a U-shaped relationship with a firm's radical innovation performance [32]. Fu, Liu, and Zhou (2019) indicated that open innovation strategies have different effects at different times. Their analysis of 172 biopharmaceutical companies' 516 annual reports reveal that openness, either as breadth or depth, has a negative impact on short-term (1–2 years) firm performance, but an inverted U-shaped curvilinear relationship will develop after about 3 years [33].

One accepted explanation for this non-consensus on the effectiveness of individual openness is "environment dependency" or "context dependency" [34,35]. For instance, in an investigation of 281 Upper Austria companies, Schweitzer, Gassmann, and Gaubinger (2011) proved that openness exerts a stronger effect on firms' innovation success in a market with higher turbulence [36]. Hung and Chou (2013) revealed that technological and market turbulence positively moderates the effect of external technology acquisition on firms' innovation performance [37]. Additionally, because of the conceptual richness of the "environment" and "context", open innovation researchers usually focus on the environmental features in one particular aspect or dimension. Together with the mentioned "turbulence", other narrow-sense environments such as "disruptiveness" [38], "network structure" [39], and "geographical proximity" [40] are also frequently discussed.

In analogy with the existing research on external knowledge search, which mainly concentrates on the effects of individual openness on individual firm performance, in this paper, we study the effects of collective openness on RIN innovation performance. Expanding Laursen and Salter's (2006) framework to the network level or collective level, we identified collective openness through the dimensions of breadth and depth as well. Collective breadth represents how widely TFs search for knowledge from external sources in general, while collective depth represents how deeply TFs draw knowledge from external sources in general. Meanwhile, we particularly focus on the influence of one type of narrow-sense environment, the disruptiveness of industrial innovation.

The disruptiveness indicates the potential of an innovation to turn an industry upside down and fundamentally change the way business operates in general. It is defined as innovation that uses developing technologies to change current performance metrics and eventually displace established competitors [41,42]. Notably, the disruptiveness integrating the technological characteristics of industrial innovation(s), i.e., radical innovation vs. incremental innovation, with productive characteristics, i.e., product innovation vs. process innovation, mirrors well the stage in which the industry has proceeded: When the disruptiveness is relatively high, industrial innovations are generally radical and product-oriented innovations; when the disruptiveness is relatively low, industrial innovations are generally incremental and process-oriented innovations; and the shift from high to low

disruptiveness marks the maturity of an industry [42–44]. In other words, relatively high disruptiveness mirrors the initial "emerging" stage; relatively moderate disruptiveness mirrors the following "developing" stage; and relatively low disruptiveness mirrors the later "mature" stage.

### 2.1.2. The Behavioral Process of External Knowledge Search

To clarify the effect of collective openness on RIN innovation performance, we should first understand the specific behavioral process of external knowledge search and how openness acts in the process. Although the behavioral process of external knowledge search is not fully explained by the existing studies [45,46], we can still trace valuable supplements of this topic in other relevant research fields, e.g., social network, inter-organizational network, bilateral games, etc. We perceive the behavioral process of external knowledge search as the process of alliance formation of the individual TF. This process includes three main steps: search boundary demarcation, partner selection and alliance decision, and alliance engagement.

(1)    Search boundary demarcation

When executing external knowledge search, a TF's first step is to discern its search scope. The external knowledge sources situated in the search scope are the candidates or potential partners. Generally, the search scope of an individual TF is determined by two factors. One is the TF's breadth, which is approximate to the "search radius" and decides the largest possible search scope. The other is knowledge complementarity, which further confines the feasible search scope of an individual TF [47,48]. When the knowledge basis of two TFs overlap too much, there is little point in knowledge sharing; when the knowledge bases of two TFs are too far apart, there is still little likelihood of knowledge sharing [49,50]. Thus, the demarcation of external search boundary is determined by TF's breadth of openness and knowledge complementarity.

(2)    Partner selection and alliance decision

After the demarcation of external knowledge search boundaries, TFs begin to select the "optimal" external knowledge sources with which to ally or collaborate. The main motivation of partner selection and alliance decision is mutual benefits and risk mitigation. On one hand, a group of scholars considers that partner selection and alliance decision are negotiated bilaterally rather than dominated unilaterally [51,52]. Mutual benefits are the preliminary for partner selection and alliance decision; i.e., an alliance or collaboration should be beneficial to all participators [53,54]. On the other hand, several scholars assert that risk mitigation should be considered in the process of partner selection and alliance decision. For instance, Kim et al. (2006) argued that firms prefer to maintain previous and current partnership because such repetition mitigates alliance risk [55]. Moreover, Rowley (1997) noted that for the sake of risk mitigation, firms are also likely to build new relationships recommended by their partners [56]. Furthermore, Rosenkopf and Padula (2008) revealed that besides collaborating with "acquaintances", firms also tend to build cluster-spanning ties to reach non-local or local peripheral knowledge sources that provide distinct valuable ideas [57].

(3)    Alliance engagement

Two critical factors should be taken into consideration in this step. One is a TF's depth, which is approximate to "digging deepness". As a TF's depth increases, the complexity the and tacit nature of the knowledge it can acquire from outside increase [58,59]. Usually, a TF with high depth tends to form strong alliance, while that with low depth tends to build weak alliance. Another factor is a TF's absorptive capacity. Absorptive capacity decides the effectiveness of a certain alliance [60,61]. A lower absorptive capacity can shrink the effect of a strong alliance, while a higher absorptive capacity can amplify the effect of a weak alliance.

(4)     Analysis framework of external knowledge search

We summarize the behavioral process of external knowledge search and construct the analysis framework as shown in Figure 1.

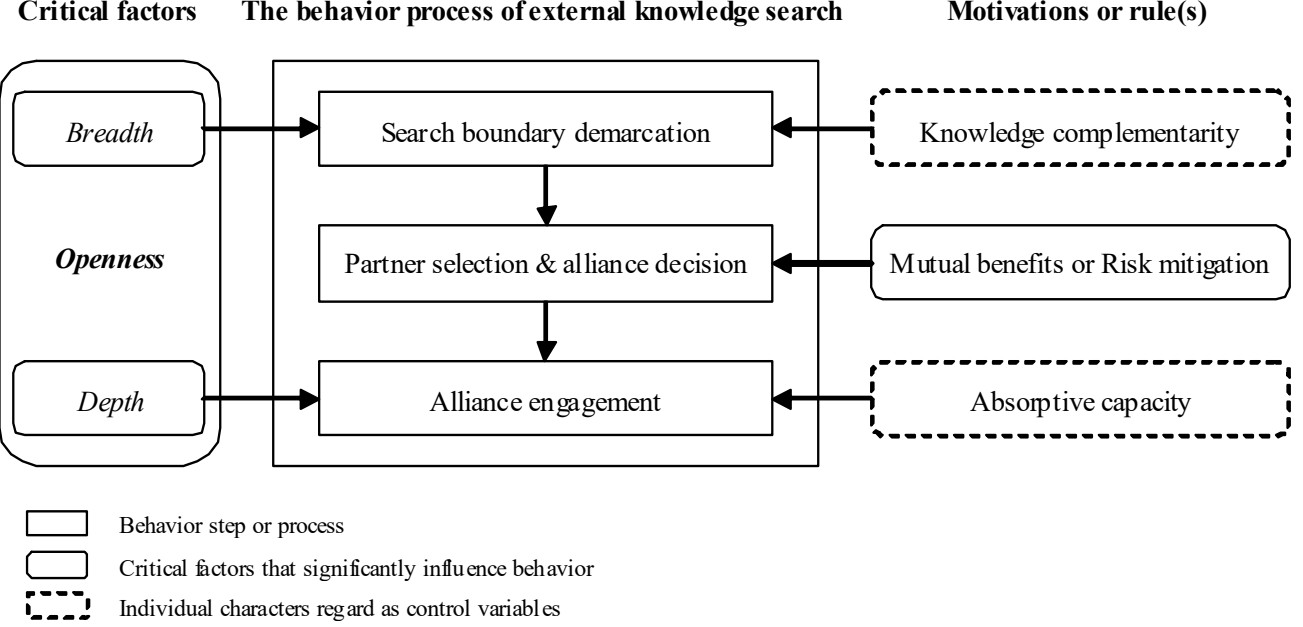

**Figure 1.** The behavioral process of an individual TF's external knowledge search.

I. There are three main steps in the behavioral process of external knowledge search: search boundary demarcation, partner selection and alliance decision, and alliance engagement.

II. Openness exerts effects in the first and third steps. Specifically, breadth together with knowledge complementarity acts in the first step, and depth together with absorptive capacity acts in the third step. We treat openness as the independent variables and knowledge complementarity and absorptive capacity as control variables.

III. Additionally, in the second step, mutual benefits and risk mitigation are the main motivations and rules of partner selection and alliance decision. To focus on the analysis of openness, we simplify that all TFs follow the "win-win" rule and have the same degree of risk tolerance.

### 2.2. RIN Innovation Performance
2.2.1. Network-Level Performance

The issue of network-level performance as an area of research is still in its infancy [22–24]. Relatively few studies on network-level performance have been carried out, and those that have been conducted apply different definitions, concepts, and measures [18,24].

For instance, Straub et al. (2004) conceptualized supply network performance through the mutual information-sharing practices of clients and vendors and measure network performance by aggregations of individual firm performance, such as operating margins, net trade cycles, and working capital efficiency [62]. Similarly, Moeller (2010) assessed business network performance by surveying accounting managers' perceptions of achieving their objectives regarding sales growth and reduction of production costs, value creation, and increase in profit [63]. Sandström and Carlsson (2008) defined the performance of policy-driven innovation network from the dimensions of efficiency and innovativeness [64]. They evaluated network efficiency by networking consequence and duration while evaluating network innovativeness through the indicators of existing educational program numbers, prevailing collaboration projects, and launched new ideas. Van der Valk, Chappin, and Gijsbers (2011) investigated innovation network performance from the perspective of net-

work structure and network resource [23]. They considered that structural features reflect a network's inner knowledge diffusion, while resources are decisive in explaining performance difference between participant firms. In accordance with knowledge-based theory, Baum et al. (2010), Tomasello (2015), and Cheng et al. (2020) introduced the notion of knowledge space (KS) to analyze the overall innovation performance of a collaborative network [44,49,65]. They considered that a collaborative network corresponds to a certain two-dimensional, abstract KS, and each participant firm can be traced by its own KS location. When innovation occurs, TFs relocate or dislocate in the KS. These relocations and dislocations, in turn, change the innovation network performance.

The relevant literature examples above show that current research on network-level performance scatters in the fields of business network, supply network, collaborative innovation network, etc. The relevant studies mainly adopt the principle that network-level performance is composed of all its participants' performance. This shows us that we can measure and analyze RIN innovation performance through the (firm) population-level performance distribution.

### 2.2.2. The Measurement

(1) Measuring indicators

Because of the similarity in research object (RINs and collaborative innovation networks) and in the research approach and method (bottom-up approach and agent-based simulation method), we refer in our research to Baum et al.'s (2010), Tomasello's (2015), and Cheng et al.'s (2020) studies [44,49,65] to conceptualize and measure innovation network performance. In the three simulation studies, the authors simplify innovation networks as "knowledge networks", where firms cooperate with each other for both knowledge learning and knowledge creation. They conceptualize that TFs are located in a multi-dimensional KS, and the location coordinates represent each firm's knowledge basis. When a TF absorbs external knowledge, it relocates within the KS. When it generates new knowledge via cooperation, it not only changes its own locations but also dislocates all other participants. These relocations and dislocations, in turn, contribute to the whole innovation network performance. Specifically, Baum et al. (2010) mainly focused on the "creation" performance calculated by the dislocations; Tomasello (2015) paid more attention to "learning" performance as the result of relocations, while Cheng et al. (2020) took both "creation" and "learning" performance into account.

In our paper, like Cheng et al., we classify innovation network performance into two components: knowledge learning performance (PKL) calculated as the relocation in the KS, and knowledge creation performance (PKC) calculated as the dislocation in the KS.

(2) Measuring pattern

The existing research on organization innovation performance is generally firm-level and immersed mainly in the average-centered view [66–68]. The average-centered view focuses on the average effects. This view assumes that the mean of innovation performance represents the central tendency of a firm's innovation outcomes and deems that the variance of innovation performance is an outlier that can be overlooked. This assumption is not appropriate to explain TF's practice in the real world. In most circumstances, TFs behave in different manners and eventually achieve different innovation performance in a certain range. The distribution of TFs' innovation performance appears non-symmetrical (usually right-skewed), which implies the likelihood of superior innovation performance achievement [66,69,70]. The variance of the distribution should be given attention in the innovation performance analysis, especially in the prediction of superior innovation outputs.

Recently, strategic management scholars have emphasized the importance of performance variance. They propose a variance-centered view to complement the average-centered view [44,67,68]. The variance-centered view focuses on the variance effects. This view concentrates on the variance of performance distribution and explains another important portion of innovation output—the occurrence of abnormal outputs. By

combining the average-centered and the variance-centered view, we can portray and examine the overall perspective of performance and make prescriptions for stimulating superior performance.

We combine the average-centered and the variance-centered view to study RIN innovation performance in our paper. We treat RIN innovation performance as the population-level innovation performance distribution, and analyze it in terms of the mean and variance of the distribution. The mean reflects an RIN's innovation capacity and innovation outcome in general. The variance indicates the proportions and possibilities of an RIN's inner ultra-conventional innovative outcomes.

*2.3. Theoretical Model*

We aim to untangle the effect of OI practice on RIN innovation performance in our paper. We adopt a bottom-up approach to analyze an RIN's innovation performance through the collective behavior of individual TFs' external search [53,71,72]. Moreover, the effectiveness of collective behavior relies on the environmental context. Hence, we develop our theoretical model, as in Figure 2.

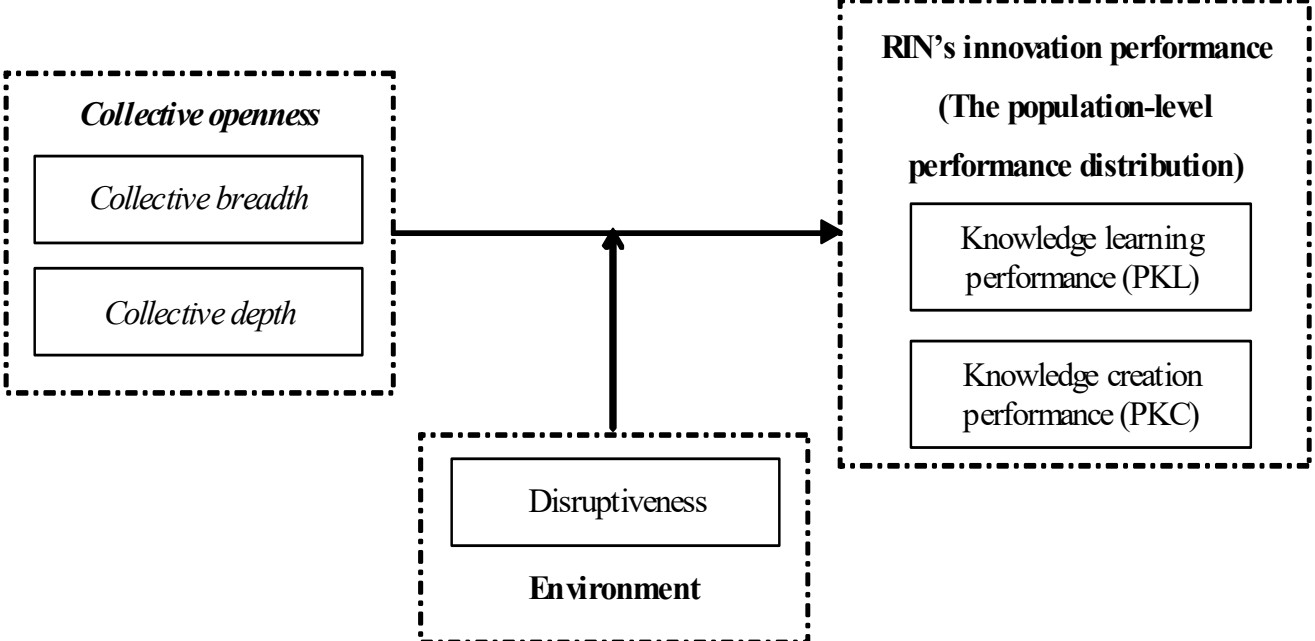

**Figure 2.** The theoretical model.

In our theoretical model, collective openness is the independent variable, the mean and the variance of RIN innovation performance are the dependent variables, and disruptiveness is the moderator variable. In analogy with the use of individual openness, we utilize collective openness to describe the behavioral characteristics of TF's collective external knowledge search. Expanding Laursen and Salter's (2006) framework to the network level or collective level, we also identify collective openness through the dimensions of breadth and depth. Collective breadth represents how widely TFs search external knowledge, in general, while collective depth represents the extent to which TFs utilize knowledge from outside in general. Meanwhile, we particularly focus on the influence of one type of narrow-sense environment, namely disruptiveness, which describes the extent of science and technology change in a certain RIN.

In particular, unlike Laursen and Salter's (2006) conceptualization and measurement of individual openness from behavioral results, we measure collective openness in accordance with behavioral process characteristics, as shown in Section 2.1.2 in our paper.

## 3. Methodology

We adopt the agent-based modeling and simulation (ABMS) approach to carry out our bottom-up study. In the bottom-up study, we think that the macro change, i.e., the emerging and evolution of an RIN, is contributed to by all the micro attributions, i.e., collective OI practice or the joint force of every individual firm's OI practice, and the RIN's environment context.

Through ABMS, we can replicate both behavioral (e.g., repeated ties) and structural (e.g., sparsely connected and locally clustered) properties of innovation networks as well as innovation outcomes [49,50,53]. When given a certain initial value of each relevant static parameter (inputs), e.g., collective breadth, collective depth, disruptiveness, etc., we can thereby collect the corresponding large-scale data generated regarding innovation performance of all individual firms (outputs) through repeated simulation experiments. The collection of the large-scale data can be used to describe the firm population performance distribution, which represents the whole RIN innovation performance. Relying on the relevant inputs and outputs, we can examine relationships between collective openness and RIN innovation performance under different disruptiveness.

### 3.1. Model Description

According to our theoretical model and the behavioral process of individual TF's external knowledge search, we designed an agent-based model to conduct our research.

(1) In our model, an RIN is defined as a complex system embedded in a certain environment within certain disruptive degree and is composed of a fixed number of TFs. All TFs are endowed the similar features with a certain extent of common collective openness to execute the external search for knowledge learning and knowledge creation. The RIN corresponds to an abstract KS, and each TF can be traced by its KS location;

(2) The external knowledge search behavior of TFs not only forms the RIN but also changes TFs' KS locations simultaneously. With the proceeding and accomplishment of TFs' external search, knowledge learning performance (PKL) and knowledge creation performance (PKC) are generated. Additionally, we assess RIN innovation performance from the aspects of the mean and the variance;

(3) TFs execute their external knowledge search under the following guidance:

- Breadth together with knowledge complementarity determines TF's external knowledge search scope. Specifically, breadth confines the possible radius of external search, while knowledge complementarity further confirms the feasible scope of external search;
- Mutual benefits and the preference of risk mitigation guide the behavior of partner selection and alliance decision in TF's external knowledge search;
- Depth together with absorptive capacity determines the engagement of alliance between TFs.

### 3.2. Simulation Model Description in Mathematics

#### 3.2.1. RIN and Knowledge Space

We abstracted our model in mathematical terms for the simulation convenience. In our simulation model, an RIN is composed by a fixed group of TFs and corresponds to an abstract, i.e., $[0, 1]^2$, two-dimensional KS metric.

Each TF embedded in the RIN can be traced by its KS location, which represents the knowledge endowment in the form of a pair of positive real numbers, $0 \leq x_i, y_i \leq 1$. Notice that a larger $x$ or $y$ does not necessarily mean more knowledge in the corresponding dimension.

The KS distance between two TFs, namely TF $i$ and $j$, is

$$d_{ij} = \sqrt{(x_j - x_i)^2 + (y_j - y_i)^2},\tag{1}$$

### 3.2.2. Calculation of RIN Innovation Performance

(1) Calculation of PKL

A successful knowledge learning event alters a TF's KS location. The event is expressed by a simple partial linear adjustment: when TF *i* learns from TF *j*, *i* will increase its similarity with *j* in terms of its knowledge basis, thus relocating its KS location according to

$$
\begin{cases}
\Delta x_i^t = \alpha \cdot \mu_i^t (x_j^t - x_i^t) \sim U(0, \alpha \left| x_j^t - x_i^t \right|) \\
\Delta y_i^t = \alpha \cdot \mu_i^t (y_j^t - y_i^t) \sim U(0, \alpha \left| y_j^t - y_i^t \right|)
\end{cases},
\tag{2}
$$

where parameter $\alpha \in (0, 0.5)$ is a constant and represents TF *i*'s absorptive capacity; $\mu_i^t$ represents TF *i*'s individual depth of openness in a certain time interval, following a uniform random distribution of U(0, $\mu$); and $\mu \in (0, 1]$ represents the collective depth.

Consequently, TF *i*'s learning performance in time interval *t*, i.e., $PKL_i^t$, is calculated as

$$
PKL_i^t = \sqrt{(\Delta x_i^t)^2 + (\Delta y_i^t)^2} = \alpha \cdot \mu_i^t \sqrt{(x_j^t - x_i^t)^2 + (y_j^t - y_i^t)^2},
\tag{3}
$$

and the expected relocation of TF *i* in time interval *t* is

$$
E[PKL_i^t] = \frac{\alpha}{2} \sqrt{(x_j^t - x_i^t)^2 + (y_j^t - y_i^t)^2}.
\tag{4}
$$

(2) Calculation of PKC

A successful knowledge creation event rearranges the whole layout of KS. The extent to which a non-creator is affected by a creation event is determined by both its proximity to the creator and the RIN's disruptiveness. We assume that, following a knowledge creation event, non-creators are dislocated in KS as a function of their distance to the creator TF *i*. Any non-creator TF *k* is displaced, and the maximum possible absolute variations of its *x*- and *y*-coordinates are

$$
\left| \Delta x_{k-\max} \right| = \left| \Delta y_{k-\max} \right| = \psi_k = \varepsilon \cdot \exp(-1/\theta) \cdot \left( 1 - \frac{1}{\sqrt{2}} d_{ik} \right),
\tag{5}
$$

where $d_{ik}$ is the KS distance between TF *i* and *k*; $\varepsilon \in (0, 1)$ is a scaling parameter that we use to control the maximum absolute variations of coordinates; $\theta \in (0, 1)$ represents the RIN's disruptiveness.

The realized variations, $\Delta_{xk}$ and $\Delta_{yk}$, respectively, follow a uniform distribution of U $(-\psi k, \psi k)$; thereby, TF *k*'s dislocation, $\sqrt{\Delta x_k^2 + \Delta y_k^2}$, is uniformly distributed in $[0, \sqrt{2}\psi_k)$. Consequently, the total realized dislocation caused by a knowledge creation event, also understood as the creator TF *i*'s creation performance in each time interval, is thus

$$
PKC_i^t = \sum_{k \neq i}^{N} \sqrt{\Delta x_k^2 + \Delta y_k^2},
\tag{6}
$$

and the total expected dislocation caused by TF *i* in time interval *t* is

$$
E[PKC_i^t] = \sum_{k \neq i}^{N} E[\sqrt{\Delta x_k^2 + \Delta y_k^2}] = \frac{\sqrt{2}}{2} \cdot \sum_{k \neq i}^{N} \psi_k.
\tag{7}
$$

### 3.2.3. TF's External Search Process

(1)　Potential partner set identification

A TF's potential partner set is determined by the knowledge complementarity and breadth of openness. The former factor determines the shortest KS distance between a pair of potential partners, while the latter determines the longest KS distance.

If TF $j$ is a potential partner of TF $i$, their distance follows as

$$d_c \leq d_{ij} \leq \rho \cdot \lambda_i^t, \tag{8}$$

where $d_c$ is the minimum external search scope; $\lambda^t{}_i$ is the TF $i$'s individual breadth of openness in time interval $t$, following the uniform random distribution U $(0, \lambda)$; $\lambda \in (0, 1]$, represents collective breadth; $\rho \in (0, 1)$ is a scaling parameter that we use to control the maximum scope of external search.

(2)　Partner selection and alliance decision

Partner selection and alliance decision are oriented as mutual benefits and risk mitigation. We treat the orientation of risk mitigation as the possibility of successful matching and further interaction between coupled TFs. We model the possibility as a single-peaked function of the KS distance between coupled TFs. Formally, we employ a bell-shaped (Gaussian) function to map one-period possibilities of an (potential) alliance, $I \leftarrow j$, according to

$$\eta_{i \leftarrow j}^t \equiv f(d_{ij}^t) = \begin{cases} 0 & d_{ij} \notin (d_c, \rho \cdot \lambda_i^t) \\ \eta' \exp\left(-(d_{ij}^t - d')^2 / \sigma^2\right) & d_{ij} \in (d_c, \rho \cdot \lambda_i^t) \end{cases}, \tag{9}$$

where $d'$ is the optimal KS distance between TF $i$ and $j$; and the positive $\eta' \ll 1$ is a scaling parameter controlling the maximum likelihood.

We set mutual benefits of the alliance between TF $i$ and $j$ as

$$\pi_i^{i \leftarrow j} = \eta_{i \leftarrow j} \cdot (E[PKL_i^t] + E[PKC_i^t]) - C\mu_i^t, \tag{10}$$

where $\eta_{i \leftarrow j}$, $E[PKLi]$ and $E[PKCi]$ were defined in Equation (4), Equation (7), and Equation (9), respectively; the positive parameter C represents the coefficient cost of building or maintaining the alliance.

(3)　Alliance engagement

The engagement of alliance is determined by TF $i$'s depth of openness and absorptive capacity. The strength of alliance $i \leftarrow j$ in time interval $t$ is

$$S_{i \leftarrow j}^t = \alpha \cdot \mu_i^t, \tag{11}$$

where parameter $\alpha$ and $\mu^t{}_i$ are the same as those in Equation (2).

### 3.3. Simulation Settings

The simulation settings are shown in Table 1. The magnitude of the simulation outputs depends on the parameters of $\lambda$, $\mu$, and $\theta$, each of which varies from 0 to 1. Parameters $\lambda$ and $\mu$ gradually increase by 0.05 from the initial value of 0.1. Therefore, given a certain value for $\theta$, there are 19 × 19 = 361 kinds of simulation experiments. For every kind of numerical experiments, we ran the simulation 25 times. At each time, the model is run for 1100 time-steps, but the first 100 are discarded to eliminate systematic errors. When the simulation experiments are run, the analytical data regarding every TF's innovation performance at every timestep is generated.

<div align="center">**Table 1.** Parameter settings and output indicators.</div>

| Items | Definition | Attribute(s) | Role in Theoretical Model | Numerical Value |
|---|---|---|---|---|
| $\lambda$ | *Collective breadth* | Parameter, Static continuous | Independent variable | Ranging from 0 to 1 and regulated by simulation input |
| $\mu$ | *Collective depth* | Parameter, Static continuous | Independent variable | Ranging from 0 to 1 and regulated by simulation input |
| $\theta$ | The disruptiveness of the RIN | Parameter, Static continuous | Moderator variable | Ranging from 0 to 1 and regulated by simulation input |
| $N$ | The number of TFs in RIN | Parameter, Constant | Control variable | 100 |
| $\alpha$ | Absorptive capacity | Parameter, Constant | Control variable | 0.05 |
| $d'$ | The optimal distance between two TFs | Parameter, Constant | Control variable | 0.03 |
| $\sigma$ | The SD of Gaussian function to map one-period cooperation rates | Parameter, Constant | Control variable | 0.025 |
| $\varepsilon$ | A positive scaling parameter to control the maximum absolute variation in firms' coordinates | Parameter, Constant | Control variable | 0.5 |
| $\eta'$ | A positive scaling parameter to control for the maximum success probability | Parameter, Constant | Control variable | 0.025 |
| $\rho$ | A positive scaling parameter to control for the maximum search radius | Parameter, Constant | Control variable | 0.3 |
| $C$ | The cost coefficient of building or maintaining a relationship | Parameter, Constant | Control variable | 0.0005 |
| *Ave-PKL* | The mean value across all TFs' entire *PKL*. TF's *PKL* in each time interval is calculated according to Equation (3). | Outputs, Dynamic | Dependent variable | — |
| *CV-PKL* | The coefficient of variation, CV, across all TFs' entire *PKL*. It is calculated as "CV% = 100×SD/mean" | Outputs, Dynamic | Dependent variable | — |
| *Ave-PKC* | The mean value across all TFs' entire *PKC*. TF's *PKC* at each time interval is calculated according to Equation (6) | Outputs, Dynamic | Dependent variable | — |
| *CV-PKC* | The coefficient of variation, CV, across all TFs' entire *PKC* calculated as "CV% = 100×SD/mean" | Outputs, Dynamic | Dependent variable | — |

Our simulation experiments aim to examine how RIN innovation performance varies responding to the changes of collective breadth ($\lambda$) and collective depth ($\mu$) in certain innovation environments that are described by disruptiveness ($\theta$). Hence, we record four types of simulation outputs: *Ave-PKL*, *CV-PKL*, *Ave-PKC*, and *CV-PKC*. Specifically, $\lambda$ and $\mu$ are ranged from 0.1 to 1, with 19 constant increments on a log-scale. $\theta$ is set at the numerical value of 0.05, 0.25, and 0.55, which, respectively, means the three typical innovation environments of poorly disruptive, moderately disruptive, and highly disruptive.

## 4. Analysis of Result

### 4.1. Effect of Collective Openness in a Poorly Disruptive Environment

Figure 3 displays the features of RIN innovation performance under different open conditions in a poorly disruptive environment. Specifically, the upper panels of Figure 3

show the features of RIN innovation performance on average, i.e., *Ave-PKL* and *Ave-PKC*, while the bottom panels of Figure 3 show those on variance, i.e., *CV-PKL* and *CV-PKC*.

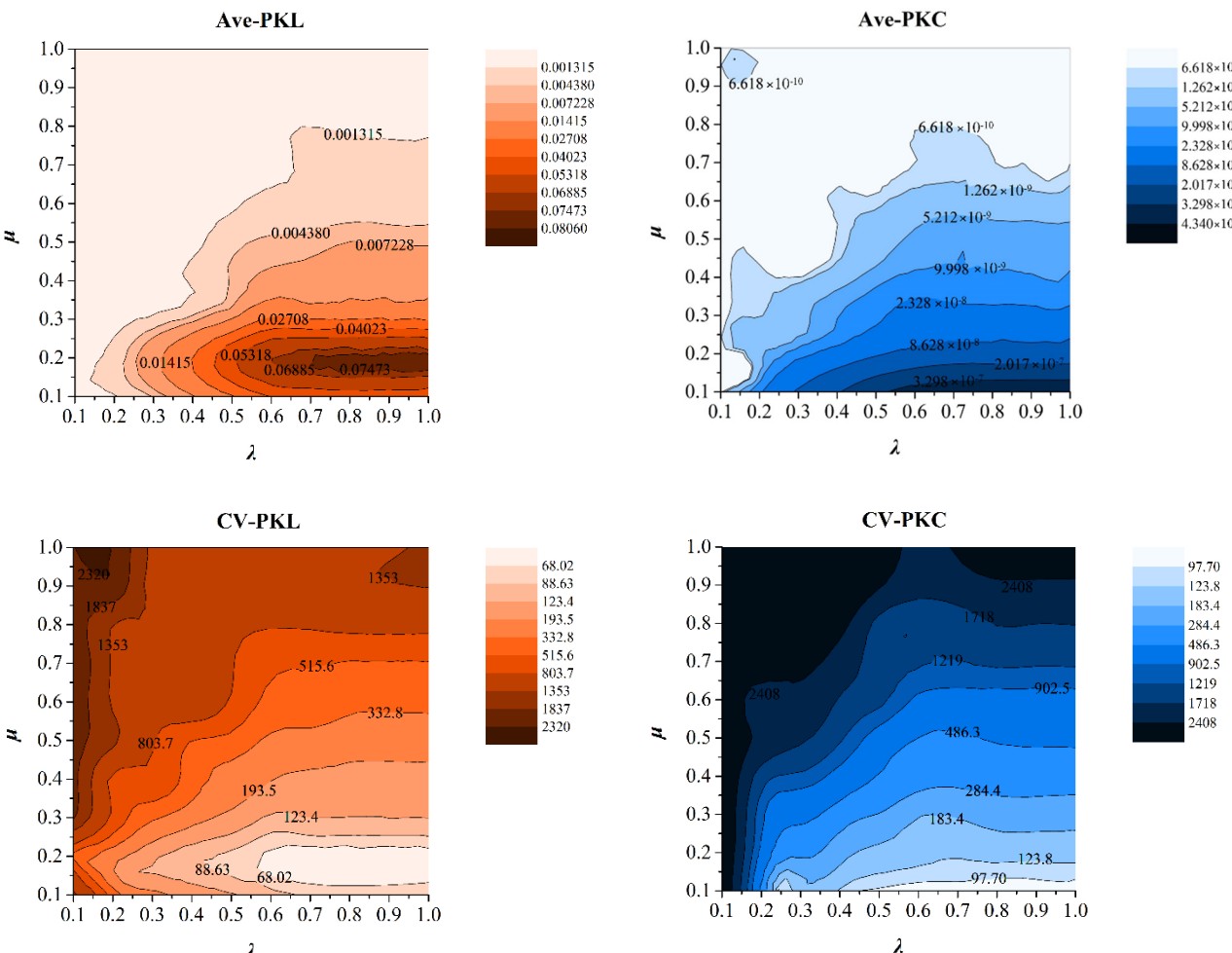

**Figure 3.** The effects of collective openness on RIN innovation performance in a poorly disruptive environment ($\theta = 0.05$).

As the upper panels of Figure 3 show, *Ave-PKL* and *Ave-PKC* are quite low in general, and the latter are far less. This means that in a poorly disruptive environment, the benefit brought by OI for an RIN is small, and the small performance improvement is mainly represented in knowledge sharing among TFs rather than new knowledge production. In most cases, the *Ave*(s) are more sensitive to the changes in collective depth than in collective breadth, as the isoclines slopes against the $\mu$-axis are generally much larger than those against the $\lambda$-axis. This implies that TFs prefer to strengthen current alliances for deep knowledge learning than build new alliances in a poorly disruptive environment.

The upper panels of Figure 3 also tell how RIN average innovation performance varies with collective openness. It can be found that the variation tendencies of the two *Ave*(s) are similar to some extent. As shown in the upper-left panel, *Ave-PKL* shows in an inverted U-shaped relationship with $\lambda$ and $\mu$, respectively. For the relationship of *Ave-PKL* and $\lambda$, the inflexion point of $\lambda$ is between 0.5 and 0.6. When $\mu$ is relatively low, that is, $\mu \in (0.1, 0.5)$, *Ave-PKL* first increases with $\lambda$ and then keeps relatively steady. When $\mu$ is relatively high, that is, greater than 0.6, *Ave-PKL* remains at a relatively low level and does not rely on $\lambda$. For the relationship of *Ave-PKL* and $\mu$, the inflexion point of $\mu$ is near 0.2. With increasing $\mu$, *Ave-PKL* rapidly increases and then gradually decreases. Additionally, as $\lambda$ increases, the inverted U-shaped relationship between *Ave-PKL* and $\mu$ becomes more prominent. As shown in the upper-right panel, *Ave-PKC* has an inverted U-shaped relationship with

$\lambda$. The inflexion point of $\lambda$ is near 0.5. As $\mu$ decreases, the relationship becomes more prominent. *Ave-PKC* is negatively influenced by $\mu$.

The bottom panels of Figure 3 indicate the variation tendencies of the two *CV*(s) responding to the changes of collective openness are similar as well. As shown in the bottom-left panel, *CV-PKL* has a U-shaped relationship with $\lambda$ and $\mu$, respectively. The inflexion points of $\lambda$ and $\mu$ are approximate to those of *Ave-PKL*, $\lambda$ is in the interval of (0.5, 0.6), and $\mu$ is approximately 0.2. For the relationship of *CV-PKL* and $\lambda$, *CV-PKL* first decreases with $\lambda$ and then keeps relative steady. For the relationship of *CV-PKL* and $\mu$, with increasing $\mu$, *CV-PKL* slightly decreases and then significantly increases. As shown in the bottom-right panel, *CV-PKC* has a U-shaped relationship with $\lambda$ and a positive relationship with $\mu$. For the relationship of *CV-PKC* and $\lambda$, *CV-PKC* first decreases and then keeps relative steady; the inflexion point of $\lambda$ is approximate to that of *Ave-PKC*, while $\lambda$ is near 0.5. For the relationship of *CV-PKC* and $\mu$, *CV-PKC* increases with $\mu$.

In summary, in a poorly disruptive environment, a limited open innovation behavior should be encouraged in the RIN. When the TFs in the RIN are in a relatively low degree of open innovation, a slight increase of collective openness will improve the RIN's innovation performance. When the TFs in the RIN are in a relatively high degree of open innovation, the increase of collective openness exerts little effects on the RIN's innovation performance but aggravates the polarization of individual TF innovation performance in the RIN.

### 4.2. Effect of Collective Openness in a Moderately Disruptive Environment

Figure 4 displays the features of RIN innovation performance under different open conditions in a moderately disruptive environment. Specifically, the upper panels of Figure 4 show the features of RIN innovation performance on average, i.e., *Ave-PKL* and *Ave-PKC*, while the bottom panels of Figure 4 show those on variance, i.e., *CV-PKL* and *CV-PKC*.

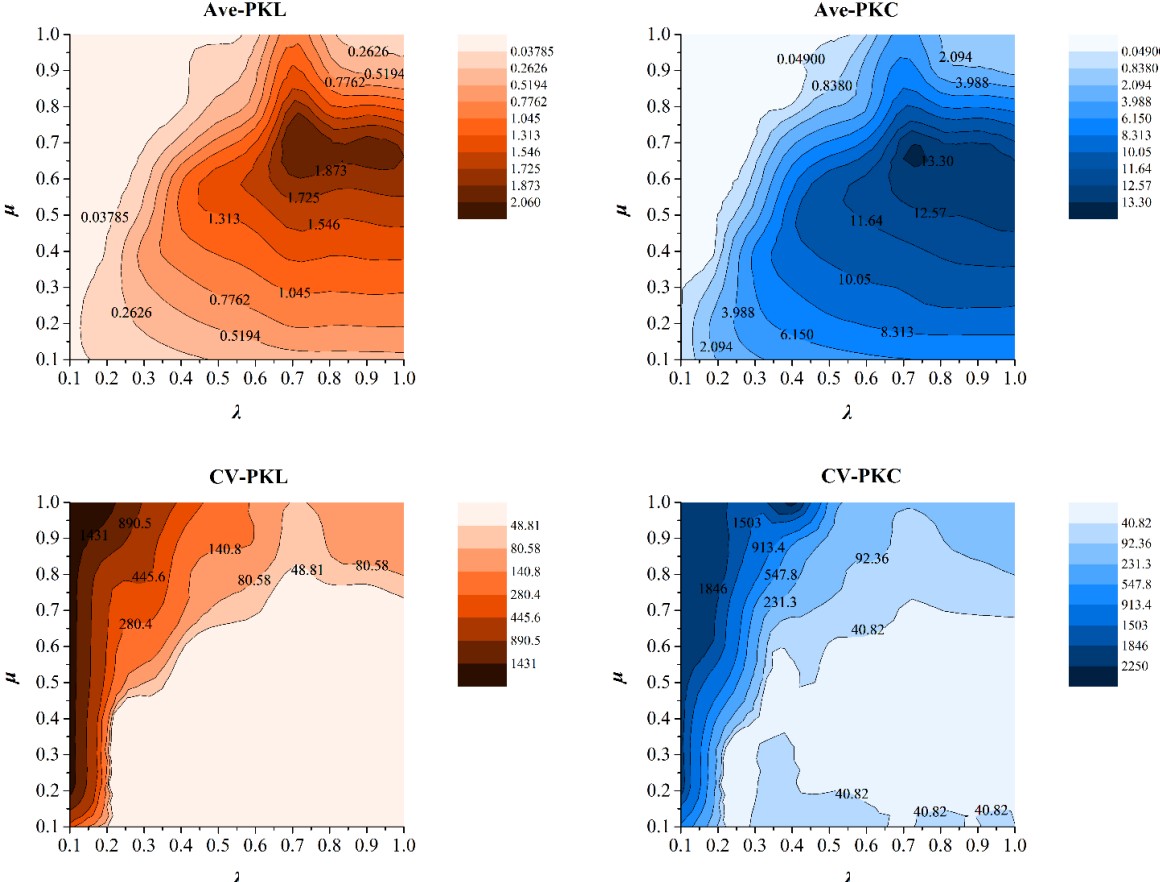

**Figure 4.** The effect of collective openness on network performance of RIN in a moderately disruptive environment ($\theta$ = 0.25).

As the upper panels of Figure 4 show, *Ave-PKL* and *Ave-PKC* are far greater than those in Figure 3, and *Ave-PKC* surpasses *Ave-PKL*. This indicates that the magnitude of collective openness' effectiveness in a moderately disruptive environment is much greater than that in a poorly disruptive environment and that knowledge creation tends to be the main way to improve RIN innovation performance. In the different regions of the $(\lambda, \mu)$ space, the *Ave*(s) appear to have different sensitivity to the changes of collective openness, reflected as the isoclines' non-monotonously changing slopes against the $\mu$-axis and the $\lambda$-axis. In the left half of $(\lambda, \mu)$, where $\lambda < 0.55$, the $\lambda$-slopes and $\mu$-slopes both fluctuate between 0 and 1; while in the right half, where $\lambda > 0.55$, the isoclines are nearly parallel with $\lambda$-axis. Thus, whether the effectiveness of collective openness is dominated by breadth or depth is dependent on an RIN's current openness level. When an RIN's breadth is relatively low, both collective breadth and collective depth play important roles in shaping the RIN's innovation performance. In contrast, when an RIN's collective breadth is relatively high, the RIN's innovation performance is influenced by more collective depth than collective breadth.

The upper panels of Figure 4 also tell how RIN average innovation performance varies with collective openness. It can be found that the variation tendencies of the two *Ave*(s) are similar to some extent. As shown in the upper-left panel, *Ave-PKL* has an inverted U-shaped relationship with $\lambda$ and with $\mu$, respectively. For the relationship of *Ave-PKL* and $\lambda$, *Ave-PKL* first increases with $\lambda$ and then remains relatively steady, and the inflexion point of $\lambda$ is near 0.7. For the relationship of *Ave-PKL* and $\mu$, with increasing $\mu$, *Ave-PKL* increases at first and then decreases, and the inflexion point of $\mu$ is near 0.6. As shown in the upper-right panel, *Ave-PKC* has an inverted U-shaped relationship with $\lambda$ and with $\mu$, respectively, as does *Ave-PKL*. The inflexion point of $\lambda$ is near 0.65, and the inflexion point of $\mu$ is near 0.6.

The bottom panels of Figure 4 show that the variation tendencies of the two *CV*(s) are similar and in a different direction compared to the *Ave*(s). As shown in the bottom-left panel, *CV-PKL* has a U-shaped relationship with $\lambda$ and with $\mu$, respectively. For the relationship of *CV-PKL* and $\lambda$, with increasing $\lambda$, *CV-PKL* first rashly decreases and then remains relatively steady. For the relationship of *CV-PKL* and $\mu$, when $\lambda > 0.2$, with increasing $\mu$, *CV-PKL* remains relatively steady at first and then significantly increases; otherwise, *CV-PKL* increases with $\mu$. The inflexion point of $\lambda$ is near 0.2, and the inflexion point of $\mu$ is between 0.45 and 0.7. As shown in the bottom-right panel, similar to *CV-PKL*, *CV-PKC* is also in a U-shaped relationship with $\lambda$ and with $\mu$. The inflexion point of $\lambda$ is near 0.2, and the inflexion point of $\mu$ is between 0.45 and 0.65.

In summary, the variation tendencies of *CV*(s) synchronize with those of *Ave*(s). These results imply that it is not necessarily true that more collective openness is better for an RIN in a moderately disruptive environment.

When an RIN's collective openness is relatively low, any increase in collective openness will significantly improve its innovation performance in general, which echoes the increasing *Ave*(s) in the bottom-left quarter of the $(\lambda, \mu)$ space. Meanwhile, the performance gaps among individual TFs in the RIN are narrowed to a certain stable level, which echoes the decreasing *CV*(s) in the bottom-left quarter of the $(\lambda, \mu)$ space.

When an RIN's collective openness is low in breadth but high in depth, the increase in collective breadth will improve its innovation performance in general, and that of collective depth will do the opposite. This finding echoes the variation tendency of *Ave*(s) in the upper-left quarter of the $(\lambda, \mu)$ space. Meanwhile, the performance gap among individual TFs in the RIN are narrowed with collective breadth increasing but are expanded with collective depth increasing, which echoes the variation tendency of *CV*(s) in the upper-left quarter of the $(\lambda, \mu)$ space.

When an RIN's collective openness is high in breadth but low in depth, the increase in collective breadth does not affect its innovation performance, while the increase of collective depth will significantly improve its innovation performance in general. This finding echoes the variation tendency of *Ave*(s) in the bottom-right quarter of the $(\lambda, \mu)$

space. Meanwhile, the performance gaps among individual TFs in the RIN are narrowed to a certain stable level as collective depth increases, which echoes the changing trend of $CV$(s) in the bottom-right quarter of the $(\lambda, \mu)$ space.

When an RIN's collective openness is relatively high, any increase in collective openness will degrade the RIN's innovation performance in general, which echoes the decreasing $Ave$(s) in the upper-right quarter of the $(\lambda, \mu)$ space. Meanwhile, the upgrade in local performance generated by a few well-performing TFs will aggravate the polarization of individual TF innovation performance in the RIN, which echoes the increasing $CV$(s) in the upper-right quarter of the $(\lambda, \mu)$ space.

### 4.3. Effect of Collective Openness in a Highly Disruptive Environment

Figure 5 displays the features of RIN innovation performance under different open conditions in a highly disruptive environment. Specifically, the upper panels of Figure 5 show the features of RIN innovation performance on average, i.e., *Ave-PKL* and *Ave-PKC*, while the bottom panels of Figure 5 show those on variance, i.e., *CV-PKL* and *CV-PKC*.

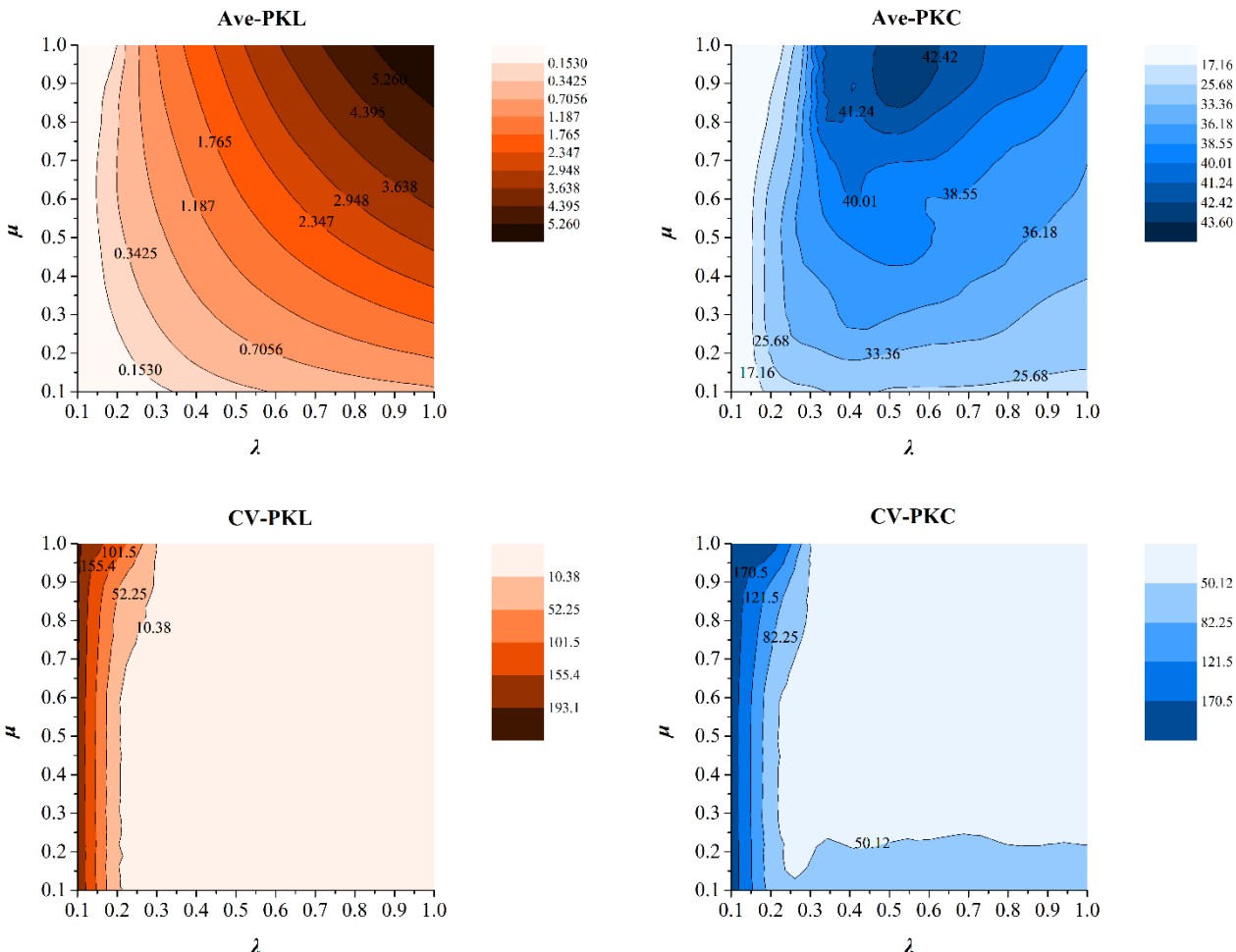

**Figure 5.** The effect of collective openness on network performance of RIN in a highly disruptive environment ($\theta = 0.55$).

As the upper panels of Figure 5 show, *Ave-PKL* and *Ave-PKC* are far greater than those in Figures 3 and 4, and *Ave-PKL* is significantly lower than *Ave-PKC*, which indicates that the magnitude of collective openness's effectiveness in a highly disruptive environment is much greater than those in the other types of environments. Knowledge creation is the main way to improve RIN innovation performance. In different regions of the $(\lambda, \mu)$ space, the *Ave*(s) presents different sensitivities to the changes of collective openness, as reflected

in the isoclines' non-monotonously changing slopes against the μ-axis and the λ-axis. In the left marginal area of the ($\lambda$, $\mu$) space, where $\lambda < 0.25$, the isoclines of the *Ave*(s) are nearly parallel with the μ-axis, and $\lambda$ is the valid and chief parameter. While in the other area of the ($\lambda$, $\mu$) space, for *Ave-PKL*, the $\lambda$-slopes and $\mu$-slopes are both approximate to 1, with both $\lambda$ and $\mu$ being chief parameters; for *Ave-PKC*, the $\lambda$-slopes are lower than the μ-slopes, with $\mu$ being the chief parameter. Thus, in a highly disruptive environment, whether the effectiveness of collective openness is dominated by breadth or depth is dependent on an RIN's current openness level.

The upper panels of Figure 5 also show the variation increases and keeps steady as collective breadth increases. This finding echoes the concave-curvilinear growth tendencies of an RIN's average innovation performance responding to the changes in collective openness. It can be found that the variation tendencies of two *Ave*(s) are dissimilar. As shown in the upper-left panel, *Ave-PKL* has a positive relationship with $\lambda$ and with $\mu$. *Ave-PKL* generally increases with $\lambda$ and with $\mu$. As shown in the upper-right panel, *Ave-PKC* has an inverted U-shaped relationship with $\lambda$ and with $\mu$. The inflexion point of $\lambda$ is near 0.65, and the inflexion point of μ is near 0.8.

However, the variation tendencies of the two *CV*(s) shown in the bottom panels of Figure 5 are similar, and they are in a different direction from both *Ave*(s). The *CV*(s) are influenced mainly by $\lambda$, as the isoclines are generally parallel with the μ-axis. With increasing $\lambda$, the *CV*(s) first rashly decrease and then remain relatively steady, and the inflexion point of $\lambda$ is near 0.25.

The variation tendencies of *CV*(s) synchronize with those of *Ave*(s). These results imply that in a highly disruptive environment, more collective openness is generally better for an RIN. Any increase of collective openness will significantly enhance an RIN's innovation performance. Specifically, the learning performance increases with collective openness in general, which echoes the increasing *Ave-PKL*. Meanwhile, the performance gaps among individual TFs in the RIN are narrowed to a certain stable level, which echoes the variation tendency of *CV-PKL*. The creation performance increases with collective depth in most case of *Ave-PKL*. Meanwhile, the polarization of individual TF innovation performance in the RIN is alleviated to some extent, which echoes the variation tendency of *CV-PKC*.

### 4.4. The Moderating Effects of Disruptiveness

We further discuss the moderating effect of disruptiveness on collective openness' effectiveness. We particularly focus on how disruptiveness moderates the relationships between collective openness and the RIN innovation performance. We thus characterize how *Ave-PKL* and *Ave-PKC* vary with collective openness and the value properties in the three typical environments analyzed in Sections 4.1–4.3. The characteristics are detailed in Table 2.

The upper part of Table 2 shows the features of *Ave-PKL*'s variation tendency in different environments. On one hand, collective breadth non-negatively influences the learning performance of an RIN, and the effectiveness varies as disruptiveness rises. In a poorly disruptive environment, where $\theta = 0.05$, *Ave-PKL* concave curvilinearly increases with $\lambda$, and the inflexion point of $\lambda$ is approximately 0.55. In a moderately disruptive environment, where $\theta = 0.25$, *Ave-PKL* concave curvilinearly increases with $\lambda$, and the inflexion point of $\lambda$ moves to the right, approximately 0.70. In a highly disruptive environment, where $\theta = 0.55$, *Ave-PKL* monotonously increases with $\lambda$, and the inflexion point of $\lambda$ vanishes in the value interval of (0, 1) or moves to the far right. Therefore, the moderating effect of this facet is rather intuitive: disruptiveness prolongs the positive effect of collective breadth on the learning performance of an RIN.

On the other hand, from the upper part of Table 2, collective depth curvilinearly influences the learning performance of an RIN, and the effectiveness varies as disruptiveness rises. In a poorly disruptive environment, where $\theta = 0.05$, *Ave-PKL* evolves in an inverted U-shaped trajectory as $\mu$ increases, and the inflexion point of $\mu$ is approximately 0.2. In a moderately disruptive environment, where $\theta = 0.25$, *Ave-PKL* also evolves in an inverted

U-shaped trajectory as $\mu$ increases, and the inflexion point of $\mu$ moves right, approximately 0.60. In a highly disruptive environment, where θ = 0.55, *Ave-PKL* monotonously increases with $\mu$, and the inflexion point of $\mu$ vanishes in the value interval of (0, 1) or moves to the far right. Therefore, the moderating effect in this aspect is rather intuitive: disruptiveness prolongs the positive effect of collective depth on the learning performance of an RIN.

**Table 2.** The variation tendencies of the network performance of an RIN responding to the changes in openness.

| | Low Disruptive Environment (θ = 0.05) | Moderate Disruptive Environment (θ = 0.25) | High Disruptive Environment (θ = 0.55) |
|---|---|---|---|
| | *Ave-PKL* | *Ave-PKL* | *Ave-PKL* |
| Breadth ($\lambda$) ↑ | ↗ → <br> The inflexion point of $\lambda$ is approximately 0.55 | ↗ → <br> The inflexion point of $\lambda$ is approximately 0.7 | ↗ <br> ——— |
| Depth ($\mu$) ↑ | ↗ ↘ <br> The inflexion point of $\mu$ is approximately 0.20 | ↗ ↘ <br> The inflexion point of $\mu$ is approximately 0.60 | ↗ <br> ——— |
| Values | (0, 0.0812) | (0.0324, 2.071) | (0.149, 5.283) |
| | *Ave-PKC* | *Ave-PKC* | *Ave-PKC* |
| Breadth ($\lambda$) ↑ | ↗ → <br> The inflexion point of $\lambda$ is approximately 0.5 | ↗ → <br> The inflexion point of $\lambda$ is approximately 0.65 | ↗ → <br> The inflexion point of $\lambda$ is approximately 0.65 |
| Depth ($\mu$) ↑ | ↘ <br> ——— | ↗ ↘ <br> The inflexion point of $\mu$ is approximately 0.60 | ↗ ↘ <br> The inflexion point of $\mu$ is approximately 0.80 |
| Values | (0, 4.344 × 10⁻⁷) | (0.0476, 13.532) | (16.86, 43.88) |

Moreover, the values of *Ave-PKL* are significantly enlarged as disruptiveness increases. Hence, besides delaying non-positive effectiveness, high disruptiveness simultaneously augments the positive effect of collective openness. We thus portray such moderating effects of disruptiveness in Figure 6.

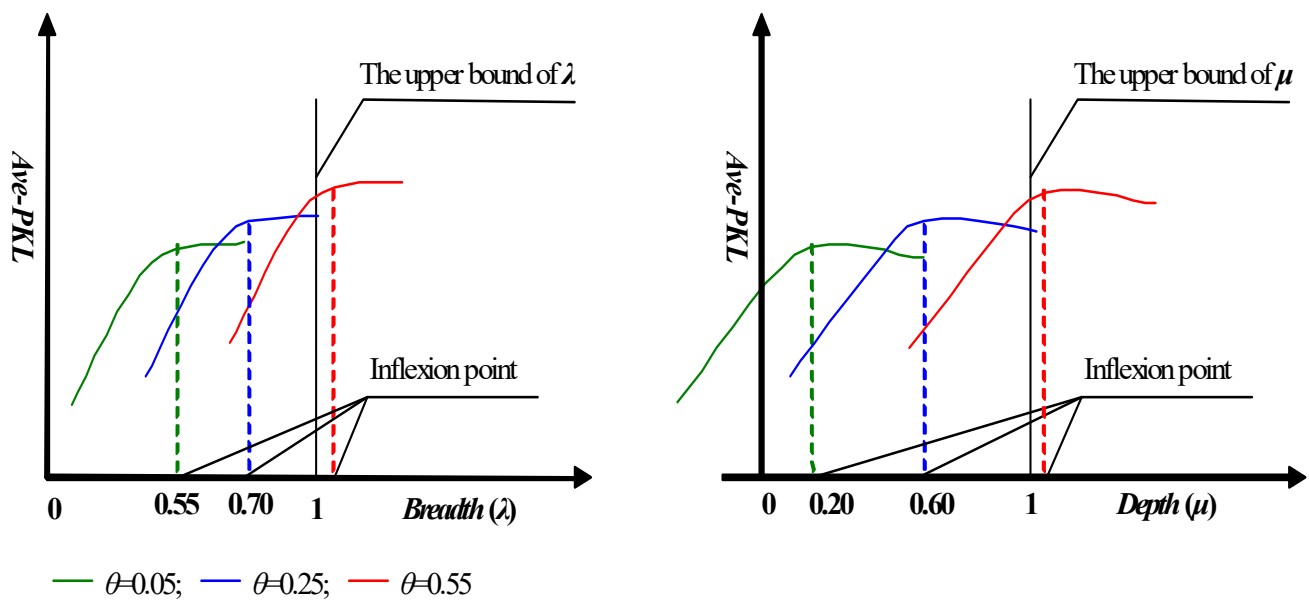

**Figure 6.** The moderating effects of disruptiveness on the effect of collective openness on the learning performance of an RIN.

The bottom part of Table 2 tells a similar story as that told by the upper part. On one hand, as disruptiveness increases, the positive effects of collective openness are significantly prolonged. In a poorly disruptive environment, the inflexion point of $\lambda$ and $\mu$ are approximately 0.5 and 0. In a moderately disruptive environment, the inflexion point of $\lambda$ and $\mu$ are approximately 0.65 and 0.60. In a highly disruptive environment, the inflexion point of $\lambda$ and $\mu$ are approximately 0.65 and 0.80. On the other hand, as disruptiveness increases, the values of *Ave-PKC* at each ($\lambda$, $\mu$) point grow dramatically. In other words, the positive effect of collective openness on the creation performance of an RIN will be significantly prolonged and augmented as disruptiveness rises.

Hence, we portray such moderating effects of disruptiveness in Figure 7.

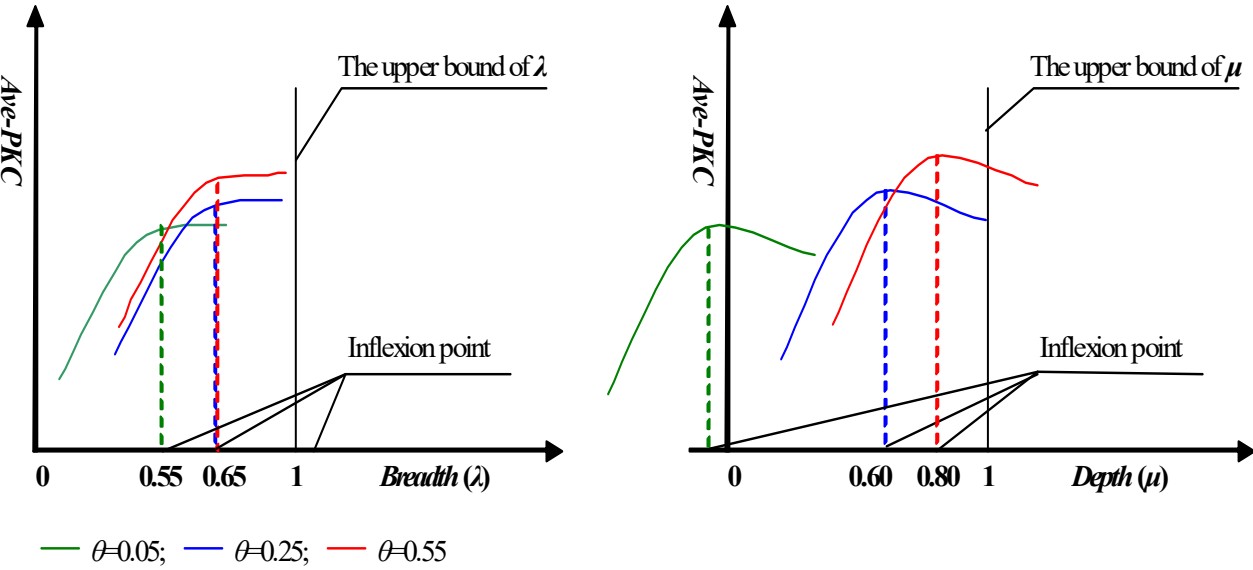

**Figure 7.** The moderating effects of disruptiveness on the effect of collective openness on the creation performance of an RIN.

## 5. Discussion

Our findings reveal that the extent to which an RIN benefits from collective openness depends on the features of the innovation environment, namely disruptiveness. Disruptiveness, depicting the extent of science and technology change, usually mirrors different stages of the industry lifecycle: high disruptiveness corresponds to the emerging stage; moderate disruptiveness corresponds to the developing stage, and low disruptiveness corresponds to the mature stage. As disruptiveness decreases, an industry advances in its lifecycle, and the positive effectiveness of collective openness is markedly shrunken and shortened to a certain extent. Such results indicate that RINs should adjust their OI strategies according to the period of industrial development. Hence, there is no innovation policy panacea for OI implementation in all RINs but rather pointed guidelines to different RINs under different environmental conditions, as shown in Figure 8.

In the emerging industrial stage, where there is much room for innovative outcomes growth, and the growth is subjected by vast potential product innovation, a proper innovation policy should pay more attention to how to fully improve RIN innovation performance. Therefore, higher level of average innovation performance across the participant TFs is the ultimate target to RINs in this period. As our simulation results show that, in this period, RIN average innovation performance is positively related with collective depth and with collective breadth in general. We thus recommend the aggressive collective openness, i.e., high collective breadth and high collective depth, which emphasizes the overall improvement of individual TFs' innovation performance in an RIN.

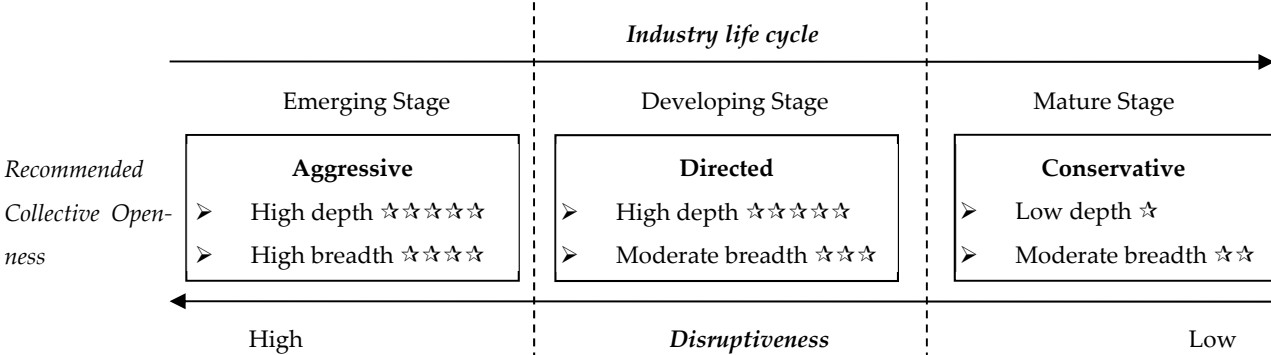

**Figure 8.** Open innovation policy prescriptions regarding collective openness for RINs.

In the developing industrial stage, where the dominant product design is formed, and the improvement of innovative outcomes is driven by the mix of product and process innovation, a proper innovation policy should focus on how to achieve superior innovation performance of RINs by supporting the few leading firms or the start-ups who have advancing knowledge and technology. Therefore, a higher level of innovation performance variance, together with the not-too-low average innovation performance, is the ultimate target for RINs in this period. As our simulation results show, in this period, for a fixed depth, RIN innovation performance variance first decreases with collective breadth and then keeps relatively stable, and for a fixed breadth, it first decreases with collective depth and then increases. In addition, RIN average innovation performance is in inverted U-shaped relationships with collective openness. We thus recommend the directed collective openness, i.e., moderate collective breadth and high collective depth, which emphasizes the fosterage of future industry leaders in an RIN.

In the mature industrial stage, the industry is routinized with relatively high outputs, but there is only a little room for innovative outcomes' increasing, and the limited increase is largely composed of process innovation. In this period, a proper innovation policy should be concerned with how to maintain the current order and status of RINs by encouraging the communication on process optimization among TFs rather than the introduction of new technologies or novel ideas. Therefore, relatively low-level performance variance, together with relatively stable-level average innovation performance, is the ultimate target for RINs. As our simulation results show, in the mature industrial stage, for a fixed-breadth RIN, innovation performance variance significantly increases with depth, and for a fixed, it gradually decreases at first and then keeps steady. Moreover, RIN average innovation performance is generally low and fluctuates around zero. We therefore recommend the conservative collective openness, i.e., moderate breadth and low depth, which emphasizes the maximum utilization of current RINs' competitive advantages.

Our suggestions are the first exploration of some OI guidelines for policymakers rather than a blueprint for OI policymaking. We hope this contributes to the growing awareness among policymakers that they have a substantial role to play in optimizing the OI benefits for RINs, and they, in this way, can contribute to the fosterage of regional innovation capacity and regional economic development.

## 6. Conclusions

In the era of knowledge explosion and globalization, regional innovation fosterage and regional economic development are increasingly relying on TFs' collective OI practice. Innovation policies should be aimed to guide individual TFs' OI practices in an RIN so as to improve the RIN's innovation performance. To deepen the understanding of how an RIN benefits from TFs' collective OI practice, we studied the relationship between collective openness and RIN innovation performance under different innovation environments in this paper. To do so, we identified *collective openness* from the dimensions of breadth and depth and analyzed how it acts in the behavioral process of external knowledge search.

Then, we studied whether the RINs that have higher openness are more likely to harvest a higher level of innovation performance in different environments. At last, we found that *collective openness* exerts significant effects on RIN innovation performance, and these effects are significantly moderated by the innovation environment, which is portrayed by disruptiveness. Our findings are shown as follows:

First, RIN innovation performance can be considerably enhanced by high collective openness, but it is not necessarily true that more collective openness is better, which reflects that the "inverted U-shaped" relationships broadly argued in firm-level open innovation studies also exist at the network level.

Second, the effect of collective openness depends on the disruptiveness of innovation environment. The likelihood of a positive effect of collective openness on RIN innovation performance increases as the disruptiveness is enhanced. The trends of the polarization of individual TFs' innovation performance in an RIN are alleviated as the disruptiveness is enhanced.

Based on these findings, we further give some guidance of innovation policymaking. When the industry is in its emerging stage, aggressive collective openness—high collective breadth and depth—aiming at achieving high RIN innovation performance is recommended. When the industry steps into its developing stage, directed collective openness—high collective depth and moderate collective breadth—aiming at fostering future industry leaders in the RIN is recommended. When the industry reaches its maturity, conservative collective openness—low collective depth and moderate collective breadth— aiming at maximum utilization of current RIN's competitive advantages is recommended.

In summary, there are four main contributions in our paper: (1) Tapping into the underexplored topic of the effects of OI at the network-level, we specifically examined the relationships between collective openness and RIN innovation performance, which enriches the current OI research that is largely composed of firm-level or company-centric studies. (2) Based on the process of external search, we established an analysis framework of individual TFs' OI behavior in a network perspective. (3) We developed an assessment pattern for network-level innovation performance, which complements both the literature of organization performance that mainly focuses on individual firm performance and the literature on inter-organization networks that mainly focus on the issues of network structure and governance hierarchy. (4) We provide some managerial implications on the practical issue of innovation policymaking for the sake of RIN innovation fosterage.

## 7. Limitations and Future Directions

The current study only explores the implications of the openness of TFs' collective external search practice for RINs. We acknowledge that the suggested guidelines need more detailed analysis in future research.

First, as we analyzed in the framework of external search behavior (shown in Figure 1), openness mainly acts in the steps of the search scope demarcation and alliance engagement, and its effectiveness is probably not only moderated by knowledge complementarity and absorptive capacity acting in the same steps but also moderated by the other important rules acting in the step of partner selection and alliance decision. However, we simplified these critical factors or rules as control variables rather than systematically examining these possible moderating effects. Hence, future research should shed light on these issues.

Second, we only focused on one narrow "environment-dependence", namely the "disruptiveness-dependence", of the effectiveness of collective openness. Actually, other important environmental constructs, e.g., turbulence, network structures, geography proximity, etc., also significantly influence the relationships between collective openness and RIN innovation performance at a high possibility. Therefore, it is necessary to examine these effects on collective openness' effectiveness and clarify the relatedness between these effects in future.

Another suggestion is to explore the implications of the openness of other typical OI practice for RINs. In reality, in an open RIN, besides the external knowledge search, there

simultaneously exist several typical OI behaviors, such as outsourcing, crowdsourcing, and licensing. The openness of these typical OI behaviors also has significant effects on RIN innovation performance, which will absolutely add useful insights of OI implementation for RINs. Thus, for future work, it is essential to uncover the effectiveness of these types of openness at the network level.

**Funding:** This research was funded by [National Natural Science Foundation of China] grant number [72104043], and by [China Postdoctoral Science Foundation] grant number [2022M710571].

**Institutional Review Board Statement:** Not applicable.

**Informed Consent Statement:** Not applicable.

**Data Availability Statement:** Not applicable.

**Conflicts of Interest:** The authors declare no conflict of interest.

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
