# Peer review of "The Effects of Open Innovation at the Network Level"

_sustainability, doi:10.3390/su142315519_

Round 1
Reviewer 1 Report
The paper deals with important topic- the influence of openness on innovation network evolution based on simulation experiments and agent based modelling however I don't understand how this topic is linked to sustainability. The authors should provide literature review and provide theoretical background for this. The paper provides very wide description of results however discussion section is poorly developed. The author needs to expand this section and provide importance of study conducted in the light of other studies in this field. `There are a lot of literature and empyrical studies conducted, therefore the new references should be added to extend scientific discussion.
Author Response
Reviewer #1
The paper deals with important topic- the influence of openness on innovation network evolution based on simulation experiments and agent based modeling however I don't understand how this topic is linked to sustainability. The authors should provide literature review and provide theoretical background for this.
Answer: We have re-written the introduction as follows. First, we point out that open innovation (OI) not only plays an important role in an individual technology-based firm’s (TF’s) survival and growth but also exerts a significant effect on a regional innovation network’s (RIN’s) evolution which reflects the sustainability of regional economic development. Then we briefly explain the significance of open innovation (OI) in the firm strategy setting and government policy making. We highlight the status quo, namely, that current OI research designates relatively little theoretical attention to governments’ innovation policy development compared with individual firms’ innovation strategy development. We identify the reason for this disparity in critical attention—the underexplored network-level OI, which is the research gap.
(1) The recent relevant literature are added or updated in the new introduction (Chapter 1) theoretical background (Chapter 2).
(2) Moreover, we refine our network-level OI study to the topic of “how collective openness influences network-level innovation performance” and describe how to conduct the study in the new introduction (Chapter 1). Consequently, Chapter 1-Introduction includes the information about methodology and results.
The paper provides very wide description of results however discussion section is poorly developed. The author needs to expand this section and provide importance of study conducted in the light of other studies in this field. `There are a lot of literature and empirical studies conducted, therefore the new references should be added to extend scientific discussion.
Answer:
(1) We refine our network-level OI study to the topic of “how collective openness influences network-level innovation performance under different disruptiveness (shown in Figure 2).”
(2) We have updated the relevant literature of openness and the external search behavior.
(3) We have supplemented the literature on network-level performance, which includes the studies by Straub et al. (2004), Moeller (2010), Sandström & Carlsson (2008), der Valk, Chappin and Gijsbers (2011), Baum et al. (2010), Tomasello (2015), and Koendjbiharie (2014). Because of the similarity in research object (i.e., regional innovation network and collaborative innovation network) and in research approach and method (i.e., the bottom-up approach and agent-based simulation method, respectively), we compare our research to studies by Baum et al. (2010), Tomasello (2015) and Cheng et al.(2020) to conceptualize and measure innovation network performance. We divide network-level innovation performance into two categories, knowledge learning performance (PKL) and knowledge creation performance (PKC).
Moreover, referring to March’s (1991), Cavarretta’s (2008), and Makino and Chan’s (2017) studies, we combine the average-centered and the variance-centered views to analyze the properties of network-level innovation performance. We treat network-level performance as the firm’s population-level performance distribution and analyze it in terms of the mean and (relative) variance. The mean reflects network-level innovation capacity in general, whereas the variance indicates the proportions and possibilities of high and low levels of innovation performance relative to the mean performance level. Additional details are provided in Section 2.2.
(4) We have re-written the Methodology. The model description now comprises two subsections, a basic description written for general readers and an abstraction expressed with mathematical language. We believe this two-part description more effectively communicates our model design.
(5) We have redesigned the model. And we also considered the moderating effect of disruptiveness on the relationships between openness and network-level performance. Additional details are provided in Sections 3 and 4. We examined the moderating effects of disruptiveness on the effectiveness of openness. We set the parameter of θ equal to 0.05, 0.25, and 0.05, values that respectively represent the poorly disruptive environment that models a mature industry, the modestly disruptive environment that models a developing industry, and the highly disruptive environment that models an emerging industry. Additional details are provided in Section 4.
(6) We have significantly revised the Discussion section based on the results of the repeat analysis. We compare our findings regarding collective openness with those regarding individual openness from previous OI studies and indicate similarities and differences. Based on our findings, we highlight managerial implications concerning the practical issue of “the extent to which OI should be encouraged by governments under different innovation circumstances to foster RIN innovation.” We consider that in the relatively low disruptive environment, the conservative collective openness, i.e., low breadth and depth, is recommended for a regional innovation network; in the modestly disruptive environment, mild collective openness, i.e., intermediate breadth and depth, is recommended; and in the highly disruptive environment, aggressive collective openness, i.e., high breadth and depth, is recommended.
Reviewer 2 Report
I have the next recommendations for the author of this article:
R1. The references from Introduction are not so actually. The most recent reference is from 2015. So, is this justified or not?
R2. Between lines 43 - 60 there is information about methodology, and about results. In the introduction, it is not recommended to include this kind of information.
R3. The whole of section 2 is based on outdated references. Please, update these references and analyze the aspects in the current context.
R4. It is not clear the methodology applied in this research. I recommend including a model of research for a good understanding of the approach. We identify remarks as "we define", "we use", "we assume", but is not so clear the approach, and why all of these.
R5. It is not clear explained what kind of data are used for measure the aspects regarding "innovation network", " firms’ partnerships", "open innovation", and "frequency or intension of inter-firm cooperation, impacts on the amount of firms’ acquired knowledge from their partners", "knowledge dynamics and performance evaluation". I understand that this is a simulation, but is necessary to explain the parameters of these experiments, how is generated the data from figure no. 2 and 3, what represent.
R6. In section 4 there are no references. Are all of these your own contribution?
R7. In section 4.2: How are measured "inter-firm knowledge transfer and knowledge sharing"?
R8. The title is not covered in the discussion and conclusion. For example, the impact is not clearly explained.
Author Response
Reviewer #2
R1. The references from Introduction are not so actually. The most recent reference is from 2015. So, is this justified or not?
R2. Between lines 43 - 60 there is information about methodology, and about results. In the introduction, it is not recommended to include this kind of information.
Answer: We have re-written the introduction as follows. First, we point out that open innovation (OI) not only plays an important role in an individual technology-based firm’s (TF’s) survival and growth but also exerts a significant effect on a regional innovation network’s (RIN’s) evolution which reflects the sustainability of regional economic development. Then we briefly explain the significance of open innovation (OI) in the firm strategy setting and government policy making. We highlight the status quo, namely, that current OI research designates relatively little theoretical attention to governments’ innovation policy development compared with individual firms’ innovation strategy development. We identify the reason for this disparity in critical attention—the underexplored network-level OI, which is the research gap.
(1) The recent relevant literature are added or updated in the new introduction (Chapter 1).
(2) Moreover, we refine our network-level OI study to the topic of “how collective openness influences network-level innovation performance” and describe how to conduct the study in the new introduction (Chapter 1). Consequently, Chapter 1-Introduction includes the information about methodology and results.
R3. The whole of section 2 is based on outdated references. Please, update these references and analyze the aspects in the current context.
Answer: We have updated the relevant reference in Chapter 2 - Theoretical background. After refining our research topic to “how collective openness influences network-level performance,” we re-organized and re-wrote Section 2 Theoretical background.
(1) We have now reviewed the literature on “the openness of external search.” We reviewed the effects of breadth and depth on firm performance and added the OI literature on context-dependence, based on which we designed our research model. Additional details are provided in Section 2.1.1.
(2) To clarify how openness functions in the process of external search, we developed an analytical framework (Figure 1), which informed the construction of the simulation model. Additional details are provided in Section 2.1.2.
(3) We have supplemented the literature on network-level performance, which includes the studies by Straub et al. (2004), Moeller (2010), Sandström & Carlsson (2008), der Valk, Chappin and Gijsbers (2011), Baum et al. (2010), Tomasello (2015), and Koendjbiharie (2014). Because of the similarity in research object (i.e., regional innovation network and collaborative innovation network) and in research approach and method (i.e., the bottom-up approach and agent-based simulation method, respectively), we compare our research to studies by Baum et al. (2010), Tomasello (2015) and Cheng et al.(2020) to conceptualize and measure innovation network performance. We divide network-level innovation performance into two categories, knowledge learning performance (PKL) and knowledge creation performance (PKC).
Moreover, referring to March’s (1991), Cavarretta’s (2008), and Makino and Chan’s (2017) studies, we combine the average-centered and the variance-centered views to analyze the properties of network-level innovation performance. We treat network-level performance as the firm’s population-level performance distribution and analyze it in terms of the mean and (relative) variance. The mean reflects network-level innovation capacity in general, whereas the variance indicates the proportions and possibilities of high and low levels of innovation performance relative to the mean performance level. Additional details are provided in Section 2.2.
R4. It is not clear the methodology applied in this research. I recommend including a model of research for a good understanding of the approach. We identify remarks as "we define", "we use", "we assume", but is not so clear the approach, and why all of these.
R5. It is not clear explained what kind of data are used for measure the aspects regarding "innovation network", " firms’ partnerships", "open innovation", and "frequency or intension of inter-firm cooperation, impacts on the amount of firms’ acquired knowledge from their partners", "knowledge dynamics and performance evaluation". I understand that this is a simulation, but is necessary to explain the parameters of these experiments, how is generated the data from figure no. 2 and 3, what represent.
Answer:
(1) We have re-written the literature of openness and external search behavior in Section 2.1. Additionally, we summarize the behavioral process of external knowledge search and construct the analysis framework as shown in Figure 1. In the framework, i) critical factors—breadth and depth, ii) behavior process of external knowledge search, iii) motivations or rules, and iv) there relationships are clearly showed.
(2) We have redesigned our simulation model and re-written Section 3 Methodology. The model description now comprises two subsections, a basic description written for general readers and an abstraction expressed with mathematical language. We believe this two-part description more effectively communicates our model design.
The data were generated from the numerical experiments that were controlled based on the parameters listed in Table 1. Additional details are provided in Section 3.3.
In our revised model, firms’ openness, also known as individual openness, follows the uniform randome distribution of (0, 2*collective openness). Additional details are provided in Section 3.2.
R6. In section 4 there are no references. Are all of these your own contribution?
R7. In section 4.2: How are measured "inter-firm knowledge transfer and knowledge sharing"?
R8. The title is not covered in the discussion and conclusion. For example, the impact is not clearly explained.
Answer:
(1) At the end of Chapter 1-Introduction, we describe how to conduct the study and subsequently specify the contributions of the paper, namely, to (i) examine the relationships between the openness of collective external search and RIN’s innovation performance; (ii) establish an analytical framework for examining individual TF’s OI behavior in the innovation network based on the process of external search; (iii) propose an assessment pattern for network-level innovation performance; and (iv) highlight managerial implications concerning the practical issue of “the extent to which OI should be encouraged by governments under different innovation circumstances to foster RIN innovation.”
(2) In this paper, we use “knowledge learning performance (PKL)” and “knowledge creation performance (PKC)” to describe “inter-firm knowledge transfer and knowledge sharing”. The details of the measurement are shown in Section 3.2.
(3) We have combined the Discussion and Conclusion sections and have added the limitations of our study at the end of the paper. We have significantly revised the Discussion section based on the results of the repeat analysis. We compare our findings regarding collective openness with those regarding individual openness from previous OI studies and indicate similarities and differences. Based on our findings, we highlight managerial implications concerning the practical issue of “the extent to which OI should be encouraged by governments under different innovation circumstances to foster RIN innovation.” We consider that in the relatively low disruptive environment, the conservative collective openness, i.e., low breadth and depth, is recommended for a regional innovation network; in the modestly disruptive environment, mild collective openness, i.e., intermediate breadth and depth, is recommended; and in the highly disruptive environment, aggressive collective openness, i.e., high breadth and depth, is recommended.
(4) We refine our network-level OI study to the topic of “how collective openness influences network-level innovation performance”. Therefore the title is changed to “The effects of open innovation at the network-level”.
Reviewer 3 Report
1. It is desirable to evaluate the impact of the assumption, that openness of firms is unified and stable over time in the model, on the results of the study.
2. Formatting of images #2&3, 4&5, 5&7 and 8&9 needs improvement.
Author Response
Reviewer 3
R1. It is desirable to evaluate the impact of the assumption, that openness of firms is unified and stable over time in the model, on the results of the study.
R2. Formatting of images #2&3, 4&5, 5&7 and 8&9 needs improvement.
Answer: We refine our network-level OI study to the topic of “how collective openness influences network-level innovation performance under different disruptiveness”. Therefore,
(1) We have updated the relevant literature of openness and the external search behavior and supplemented the literature on network-level performance.
(2) We have redesigned our simulation model and re-written Section 3 Methodology. The model description now comprises two subsections, a basic description written for general readers and an abstraction expressed with mathematical language. We believe this two-part description more effectively communicates our model design.
The data were generated from the numerical experiments that were controlled based on the parameters listed in Table 1. Additional details are provided in Section 3.3.
In our revised model, firms’ openness, also known as individual openness, follows the uniform random distribution of (0, 2﹡collective openness) and is stable over time. Additional details are provided in Section 3.2.
(3) The new figures and tables of simulation results, i.e. Figure 3, Figure 4, Figure 5, Figure 6, Figure 7 and Table 2 are carefully edited.
Round 2
Reviewer 1 Report
The authors did big job and addressed all comments of reviewers. The answers to reviewers comments are provided satisfying me. I do not have more comments and think that the paper can be published in current form.
Author Response
First, I must thank you for your twice critical feedback on my manuscript (sustainability-2016818). According to your comments, I significantly improved my manuscript. The comments also provided intriguing ideas for my future simulation research.
Second, I must thank you for your recommendation for publication. This endows me (especially me a non-English language researcher) with courage to do future research for international publication.
Reviewer 2 Report
Thank you for your detailed answer!
After a careful review, I have the next recommendations:
1. Please, set up an objective of this study at the end of the Introduction. The paragraph between 64-77 is too complicated. Formulate the objective only!
2. It is not appropriate to present the results Introduction. For this, we have discussions and conclusions sections.
3. Please, analyze the previous studies in the field, if there are any!
4. Regarding section 3 of the article, it is unclear how the data for simulation are obtained. Please, explain in a paragraph very clearly this aspect! Explain very clearly what means our bottom-up study!
5. Please, separate section 5 into two different sections: Discussions and Conclusions.
6. There are some formal references in APA style. Please, update these in accordance with MDPI requirements!
Author Response
r1. Please, set up an objective of this study at the end of the Introduction. The paragraph between 64-77 is too complicated. Formulate the objective only!
Answer:
In the new manuscript, the third paragraph of Part 1-Introduction (paragraph between 64-77) have been revised according to your suggestion. That is——【In this paper, we aim to open the black box of OI effects at the network-level. Concretely, we fouse on the relationship between collective OI practice and RIN innovation performance and determine how it is moderated by the disruptiveness of industrial innovation which is an important narrow-sense of RIN environment. First, we confine OI practice to the typical OI behavior—external search for knowledge, and use the notion of “collective openness” to descripe collective OI practice as well as the research of individual firm OI does.……】
r2. It is not appropriate to present the results Introduction. For this, we have discussions and conclusions sections.
r5. Please, separate section 5 into two different sections: Discussions and Conclusions.
Answer:
In the new manuscript, the paragraph (the original fourth) about the contributon has been conceled from Section 1- Introduction and added in the end of Part 6-Conclusion.
The original Section 5- Discussion and Conclusion has been separated into two different section, i.e., the new Section 5-Discussion and Section 6-Conclusion.
r3. Please, analyze the previous studies in the field, if there are any!
Answer:
There are few (whole) network-level analysis of OI. The majority of OI studies are firm-level or company-centric, e.g., focusing on how individual TFs’ OI strategies and practices influence their own or ego-network innovation performance. Consequently, prescriptions drawn from these works are suitable for individual TFs, but the availability and applicability to RINs are not guaranteed.
I think the oringinal wiritten is suitable.
r4. Regarding section 3 of the article, it is unclear how the data for simulation are obtained. Please, explain in a paragraph very clearly this aspect! Explain very clearly what means our bottom-up study!
Answer:
In the new manuscript, I have re-written the beginning of Section 3-Methodolog.
First, add a sentence to describ what is the bottom-up study.
Second, add a paragraph to explain how the data is generated. That is the new second paragraph in the Section 3-Methodology——【Through ABMS, we can replicate both behavioral (e.g., repeated ties) and structural (e.g., sparsely connected and locally clustered) properties of innovation networks, as well as innovation outcomes [49-50, 53]. When given a certain initial value of each relevant static parameter (inputs), e.g. collective breadth, collective depth, disruptiveness, etc., we can thereby collect the corresponding large-scale data generated regarding innovation performance of all individual firms (outputs) through repeated simulation expeirments. The collection of the large-scale data can be used to to describe the firm poupulation performance distritibution, which represents the whole RIN innovation performance. Rely on the relevant imputs and outputs, we can examine relationships between collective openness and RIN innovation performance under different disruptiveness.】
r6. There are some formal references in APA style. Please, update these in accordance with MDPI requirements!
Answer:
In the new manuscript, I have updated the references in accordance with MDPI requirements.
Reviewer 3 Report
Improvements are adequate.
Author Response
Reviewer #3:
Improvements are adequate.
Response:
I must thank you for your twice critical feedback on my manuscript (sustainability-2016818). According to your comments, I significantly improved my manuscript. The comments also provided intriguing ideas for my future simulation research.